# Seismic Reduction Mechanism and Engineering Application of Paste Backfilling Mining in Deep Rock Burst Mines

**Jiazhuo Li [1,2,*], Songyue Li [1], Wentao Ren [3], Hui Liu [3], Shun Liu [1] and Kangxing Yan [1]**

1   School of Mining Engineering, Anhui University of Science and Technology, Huainan 232001, China
2   The State Key Laboratory of Coal Resources and Safe Mining, China University of Mining and Technology, Xuzhou 221116, China
3   Shandong Energy Group Luxi Mining Co., Ltd., Heze 274700, China
*   Correspondence: jiazhuoli@aust.edu.cn; Tel.: +86-0554-6671160

**Abstract:** In the process of coal resources development, a large number of strip coal pillars have been left behind in the coal mines in central–eastern China. With the increase in coal mining depth year by year, the rock burst threat of strip coal pillars is becoming more and more prominent due to the influence of buried depth, geological structure, gob and other factors. Backfilling mining is the main means to recover the residual strip coal pillar. In order to investigate the effect of backfilling mining on the prevention and control of rock burst, taking the paste backfilling workface 1# of Gucheng coal mine as the engineering background, a comprehensive research method of theoretical analysis, numerical simulation and field monitoring was used to study the evolution of stress and of the overburden spatial structure of the backfilling workface under the control of the backfilled ratio. The results showed that the backfilling mining controls the movement and deformation of overburden by reducing the activity range of roof strata. The overburden fracture development height decreases with the increase in backfilled ratio, but there is a boundary effect influenced by the roof deflection before backfilling and the defective distance of roof contact. With the increase in backfilled ratio, the concentration coefficient of front abutment pressure, the vertical displacement of the roof and the development height of the plastic zone of overlying strata decreased obviously, which indicates that filling mining can effectively control the stress of surrounding rock and the movement of overlying strata. The field monitoring data showed that the influence range of the front abutment pressure of the paste backfilling workface was about 90 m and the maximum stress of the surrounding rock of the two entries did not exceed 7 MPa. The average daily frequency of microseism was 1.34, and the average daily total energy of microseism was $1.80 + 10^3$ J, which decreased by 69% and 90%, respectively, compared with the caving method working face with similar geological conditions. The data above showed that the backfilling mining can effectively reduce the working face stress level and dynamic load strength to achieve the effect of prevention and control of rock burst.

**Keywords:** strip coal pillar; backfilling mining; overburden spatial structure; ground pressure behavior

## 1. Introduction

With the large-scale mining of coal in the past 70 years, the coal mines in central and eastern China are facing the serious problem of coal mining under buildings (constructions), railways and water bodies. The shallow mineral resources are also gradually depleted, and the resource development continues to move towards the deep earth. The overall mining depth extends at an average rate of 5–10 m per year; thus, coal mining into a depth of one kilometer is about to become the new normal condition [1–3]. In order to control the surface subsidence and movement of overburden, many mines adopted the strip mining method in the early stage of construction. With the increase in mining depth, intensity and overall stress, the geological conditions become more complex and the risk of rock burst

becomes more serious [4]. As an important part of "green mining", backfilling mining can effectively protect the surface buildings (structures) and ecological environment, which is a revolutionary innovation of mining technology. A large number of practices have proved that backfilling mining can effectively control the ground pressure behavior and prevent rock burst [5]. It is of great significance to control surface subsidence, reduce the risk of rock burst, recover the remaining strip coal pillars and extend the service life of the mine by studying the law of overburden movement and ground pressure behavior in backfilling mining.

Domestic and foreign scholars have conducted in-depth research on backfilling mining. Miao et al. [6] established a structural model of rock movement under dense backfilling mining and gave a formula for calculating strata movement and predicting surface subsidence. Zhang [7] proposed the theory of equivalent mining height and revealed the mechanical mechanism of backfilling mining weakening the ground pressure behavior; Chang [8] obtained the destruction pattern of overburden under different backfilled ratios through the combination of similar and numerical simulations and analyzed the movement and deformation process of roof strata and the distribution characteristics of abutment pressure. Zhou et al. [9] proposed taking the backfilled ratio as a technical measure index to evaluate the backfilling effect. Zhang et al. [10] defined the critical backfilled ratio and established a mechanical model for solving the critical backfilled ratio with layer-by-layer breaking of different overlying rocks. Xu et al. [11] established a mechanical model of elastic foundation continuous beam and obtained the relation between the fracture development height and the compression modulus of the backfilling body. Cheng et al. [12] studied the ground pressure behavior in paste backfilling workface by numerical simulation and field monitoring, and obtained the front abutment pressure behavior and variation in maximum roof vertical displacement under different backfill intervals. Ghirian and Fall [13] studied the evolution of coupled thermal, hydraulic, mechanical and chemical properties of paste backfilling materials by means of experiments with insulated and undrained high columns. Hassani et al. [14] established a model of complex boundary conditions under narrow vein mining and simulated the backfill–rock interaction with nonlinear mechanical behavior. To reduce the cost of backfilling coal mining, Du et al. [15,16] proposed a new backfilling mining method with a low backfilled ratio called constructional backfilling coal mining (CBCM). The control effect of CBCM on overlying strata was analyzed and verified by various means, and good engineering and economic effects were obtained for engineering applications. Wang et al. [17,18] explored the effects of collector mixtures on the flotation kinetics of low-rank coal (LRC) particles by various means, and presented an improved mixture of triethanolamine oleic acid monoester and kerosene that can aid the flotation performance of LRC. Rao et al. [19] proposed a coal gangue image classification method based on machine vision, which successfully realized coal gangue classification based on feature extraction and a convolutional neural network. Qi and Fourie [20] summarized the design process of cemented paste backfill (CPB) technology, evaluated the potential applications and predicted the future research of CPB.

Taking paste backfilling workface 1# of Gucheng coal mine as the engineering background, this work studied the behavior of overburden movement and mine pressure of the paste backfilling workface in the area of strip coal pillars by means of theoretical analysis, numerical simulation, on-site microseismic and mineral pressure monitoring. This work explored the prevention and control effect of paste backfilling mining on rock burst, and the research results can provide technical guidance and theoretical support for improving the safety of deep mining and the recovery rate of coal resources.

## 2. The Inhibition Principle of Backfilling Mining on Overburden Fracture

### 2.1. The Principle of Controlling Overburden Movement by Backfilling Mining

As the working face advances, the backfilling mining fills the gob with paste. The backfilling body supports the immediate roof before it collapses, which reduces the activity range of the roof strata and is equivalent to reducing the mining height. The remaining roof

subsidence is called "equivalent mining height" [7]. With the change in backfilled ratio, the equivalent mining height also changes. In general, the damage degree of overburden is directly related to the substance of overlying strata. The increase in the deformation degree of overlying strata will lead to the increase in the possibility of fracture, and the deformation degree of overlying strata is directly controlled by the backfilled ratio. The failure of overlying strata under different backfilled ratios is shown in Figure 1 [10].

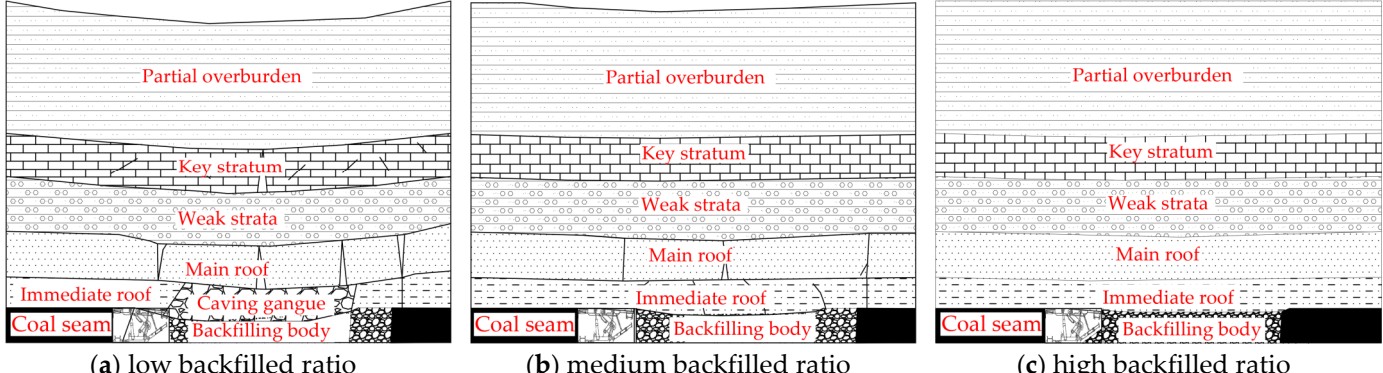

(**a**) low backfilled ratio      (**b**) medium backfilled ratio      (**c**) high backfilled ratio

**Figure 1.** Schematic diagram of overburden fracture under different backfilled ratios.

As can be seen from Figure 1, with the increase in backfilled ratio, the activity range of the overlying strata gradually decreases and the failure mode of overlying strata gradually changes from collapse to bending.

Therefore, controlling the movement and deformation of the lower roof strata is the key to controlling the overburden failure and roof pressure of the backfilling workface, and improving the backfilled ratio is the most important method to control the movement and deformation of the roof strata.

### 2.2. The Prediction of Overburden Fracture Height in Backfilling Mining under the Control of Backfilled Ratio

Xu et al. regarded the thin bedrock under thick alluvium as a bearing structure above the solid filling materials, and built the model of Winkler elastic foundation continuous beam for the single thin bedrock [11]. Through mechanical analysis, the relationship between the height of fracture development of the rock beam and the backfilled ratio is obtained:

$$
\begin{cases}
H_0' = \dfrac{\sigma_t n m H^2}{12\beta^2 qF} \times e^{\frac{m \times \eta \times h}{h - h_t - h_x}} \\
\beta = \sqrt[4]{\dfrac{k}{4EI}} \\
F = A\cos\beta x \cosh\beta x + B(\sin\beta x \cosh\beta x \\
\quad + \cos\beta x \sinh\beta x) - 2\sin\beta x \sinh\beta x
\end{cases}
\tag{1}
$$

where $H_0'$ is the fracture development height of overlying strata; $\sigma_t$ is the tensile strength of the rock beam; $n$ and $m$ are regression coefficients, which vary with the lithology and bulk of the backfill material, and can be obtained from the corresponding backfill material compression experiments; $H$ is the thickness of the backfilling body; $q$ is the uniform load of the overlying strata on the rock beam; $h$ is the height of the coal seam; $h_t$ is the roof deflection before backfilling; $h_x$ is the defective distance of the roof contact of the backfilling body; $EI$ is the flexural stiffness of the rock–beam interface; and $k$ is the foundation coefficient.

Since the complexity of the boundary conditions and the nonlinearity of the mechanical parameters of the overlying strata lead to great difficulties in the actual calculation process, the fracture zone development height can be estimated according to Equation (2), based

on an empirical formula similar to Equation (1), combined with the actual geological conditions of the backfilling workface, as shown by Xu et al. [11].

$$H_0' = \frac{100 \sum M}{1.6 \sum M + 3.6} \pm 5.6 = \frac{100(1 - \eta) \times M'}{1.6(1 - \eta) \times M' + 3.6} \pm 5.6 \tag{2}$$

where $\sum M$ is the accumulated mining height, which is the equivalent mining height in the backfilling workface, and $M'$ is the design mining height.

## 3. Overburden Fracture Development Height Prediction of the Backfilling Workface

### 3.1. Overview of the Mine and the Paste Backfilling Workface 1#

Gucheng coal mine is located in the eastern suburbs of Yanzhou District, Jining City, Shandong Province, China, with a broad and gently syncline structure. The stratum tends to the southeast and there are short axis folds in some areas. The mine is developed in a multi-level development method of vertical shaft and inclined shaft. The mine has many buildings and roads on the surface, and the Sihe River flows through the mine area. It is a typical coal mine under buildings (constructions), railways and water bodies. The mine is buried at a depth of nearly 1000 m with a complex geological structure, which was identified as a rock burst mine. In order to control the risk of rock burst, protect surface buildings and control the subsidence of overlying strata, the whole mine adopts strip mining. By 2022, there were 43 remnant pillars in the first and second levels of the mine.

The paste backfilling workface 1# is located in the south wing of district 11 in Gucheng coal mine; the east of the panel 1# is adjacent to the completed goaf 11,021, and to the west are the completed goafs 11,031, 11,051 and 11,071. To the south of the panel is the technical boundary of the mine field, and to the north is the 505 belt dip roadway. The layout of paste backfilling workface is shown in Figure 2.

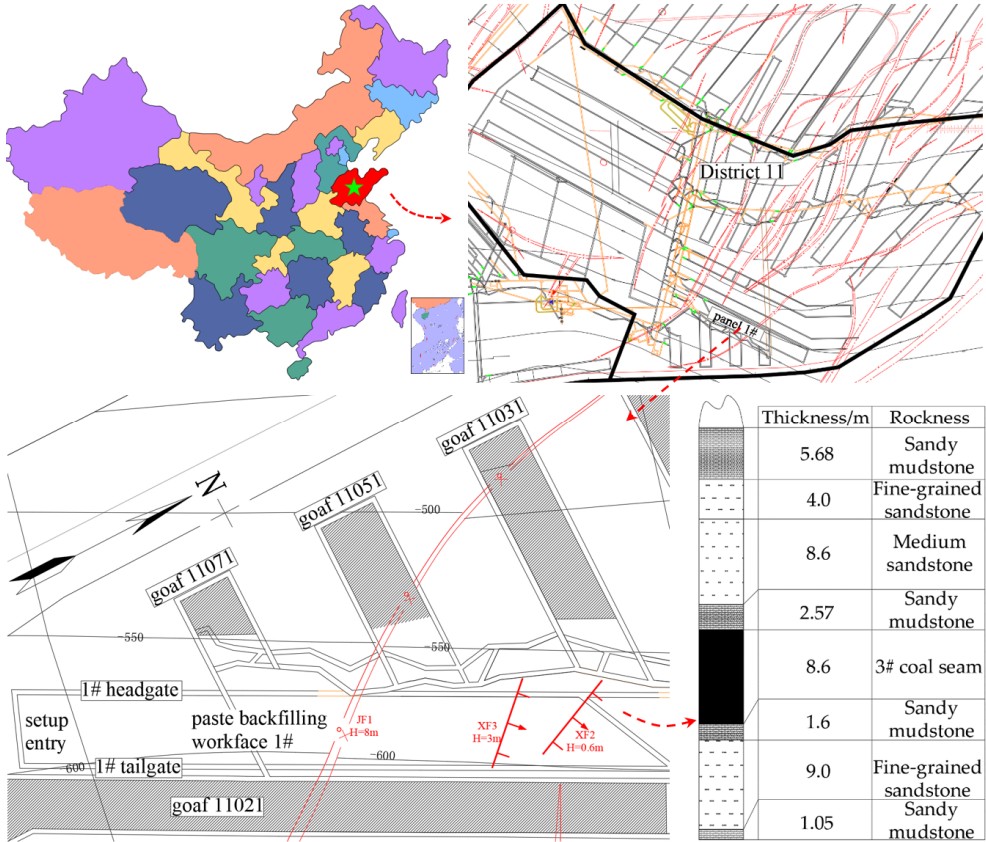

**Figure 2.** Layout of paste backfilling workface 1#.

The panel 1# is 230 m long, 55 m wide and 620–665 m deep. The area in which the panel 1# is located is an irregular shape coal pillar with a minimum width of 103 m and an unmined width of 48 m. The ratio of mined to unmined width is 55:48. In addition, the width of goaf 11,021 adjacent to panel 1# is 48 m and there was a 5 m wide barrier pillar between them; the ratio of panel 1# to the width of goaf 11,021 is 55:48.

The average coal thickness of the panel 1# is 8.6 m and the mining height is 2.8 m. In addition, the panel 1# is backfilled once every three times of coal cutting, with a backfilling interval of 2.4 m. The panel is mined along the bottom of the coal floor. The design of the roof-to-floor convergence of the panel 1# is less than 150 mm, while the backfilled ratio is not less than 96%.

### 3.2. Selection of Backfilling Materials

Backfilling materials are mainly divided into dry filling materials and wet filling materials. Dry filling materials are mainly composed of gangue and are used in Mount Lyell copper mine, Tasmania, Australia, and some coal mines in the central–eastern part of China [21]. Wet filling materials mainly include water–sand filling materials, paste filling materials, etc. The former is widely used in Polish mines [22,23] and the latter is widely adopted by coal mines in China.

After comparing with backfilling mining cases in other areas and considering the physical properties and economic effects of filling materials, Gucheng coal mine adopted paste filling materials for filling mining. The backfilling materials are divided into mortar and gangue slurry, of which the proportion of mortar is cement: fly ash: water = 200:750:800, while gangue slurry is cement: fly ash: gangue: water = 200:200:1000:460. The uniaxial compressive strength of the backfilling material is not less than 5 MPa after complete solidification.

### 3.3. Prediction of Overburden Fracture Development Height

According to Equation (2) and the actual mining technical conditions of the panel 1#, the mining height is 2.8 m, and the relationship between the backfilled ratio and the development height of the overburden can be calculated, as shown in Figure 3.

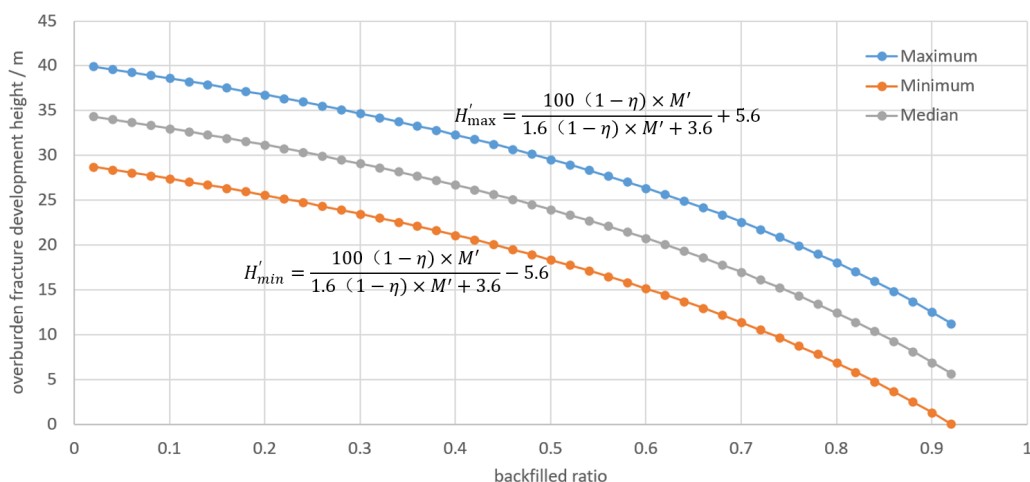

$$H'_{\max} = \frac{100\,(1-\eta)\times M'}{1.6\,(1-\eta)\times M' + 3.6} + 5.6$$

$$H'_{min} = \frac{100\,(1-\eta)\times M'}{1.6\,(1-\eta)\times M' + 3.6} - 5.6$$

**Figure 3.** Relationship between overburden fracture development height and backfilled ratio.

The median curve was taken as the estimation curve of the fracture development height of the paste backfilling workface. It can be seen from the diagram that the fracture development height of the overlying strata of the goaf decreased with the increase in the backfilled ratio and there is a boundary effect. Due to the roof deflection before backfilling and the defective distance of roof contact, the backfilled ratio of the panel cannot reach 100%. Therefore, there must be fracture development in the overlying strata.

According to the actual geological conditions of the paste backfilling workface 1# of Gucheng Coal Mine, the physical and mechanical parameters were measured by drilling and coring the roof of the panel, as shown in Table 1.

**Table 1.** Physical and mechanical parameters of overlying strata.

| Rockiness | Thickness/m |
|---|---|
| Top coal | 5.8 |
| Sandy mudstone | 2.57 |
| Medium-grained sandstone | 8.6 |
| Fine-grained sandstone | 4 |

According to Equation (2) and Table 1, the following calculations can be made:

When the first layer of the overlying strata (top coal) appeared at the cut-through fracture, i.e., the fracture development height was 5.8 m, the backfilled ratio was 91.8%.

When the second layer of the overlying strata (sandy mudstone) appeared at the cut-through fracture, i.e., the fracture development height = 8.37 m, the backfilled ratio was 87.6%.

When the third layer of the overlying strata (medium-grained sandstone) appeared at the cut-through fracture, i.e., the fracture development height = 16.97 m, the backfilled ratio was 70.1%.

When the forth layer of the overlying strata (fine-grained sandstone) appeared at the cut-through fracture, i.e., the fracture development height = 20.97 m, the backfilled ratio was 59.4% and the relationship between the backfilled ratio and the fracture of each rock layer is shown in Figure 4.

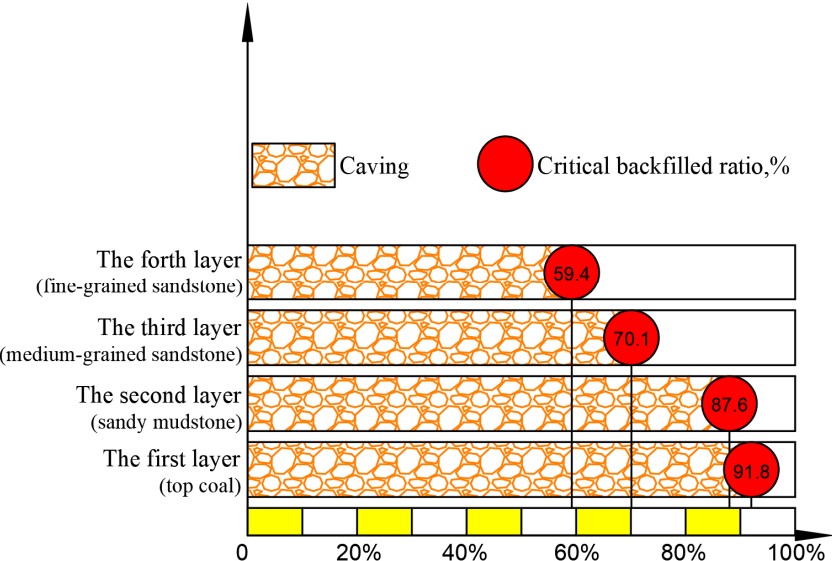

**Figure 4.** Schematic diagram of the relationship between backfilled ratio and breakage of each rock layer.

Since the backfilled ratio of the paste backfilling workface 1# was not less than 96%, combined with the aforementioned calculation, the height of the fracture development of the overlying strata was estimated to be 2.96 m, and the average sinking distance of the overlying strata was 0.11 m. The fissure could not penetrate the first layer of the overlying strata (top coal), while for the sandy mudstone layer, the overlying strata maintained good integrity, and the overall deformation was small.

## 4. Numerical Simulation of Stress and Overburden Spatial Structure Evolution of the Panel

### 4.1. Numerical Modeling

According to the borehole columnar section near the paste backfilling workface 1#, the 'Mohr–Coulomb' model was established by Flac3D software and numerical simulation was carried out. The physical and mechanical parameters of each rock layer (except the area of goaf 11021) in the model are shown in Table 2.

**Table 2.** Physical and mechanical parameters applied to numerical simulation.

| Rockiness | Friction Angle/(°) | Rock Density/(kg/m³) | Shear Modulus/GPa | Bulk Modulus/GPa | Cohesive Force /MPa |
|---|---|---|---|---|---|
| Siltstone | 36 | 2450 | 4 | 9.39 | 2.5 |
| Medium-grained sandstone | 36 | 2900 | 5 | 9 | 5 |
| Coal seam | 30 | 1300 | 2.49 | 2.67 | 2.8 |
| Sandy mudstone | 35 | 2570 | 2.4 | 3.8 | 2.5 |
| Fine-grained sandstone | 33 | 2800 | 4 | 7.07 | 4 |

The goaf 11021, which was finished in 2001, was on the east side of the paste backfilling workface 1# with a 5 m wide barrier pillar between the panels. The goaf 11031, goaf 11051 and goaf 11071, which were on the west side of the paste backfilling workface 1#, were not considered in this simulation because of the long distance from panel 1#. The layout of the simulated panel is shown in Figure 5.

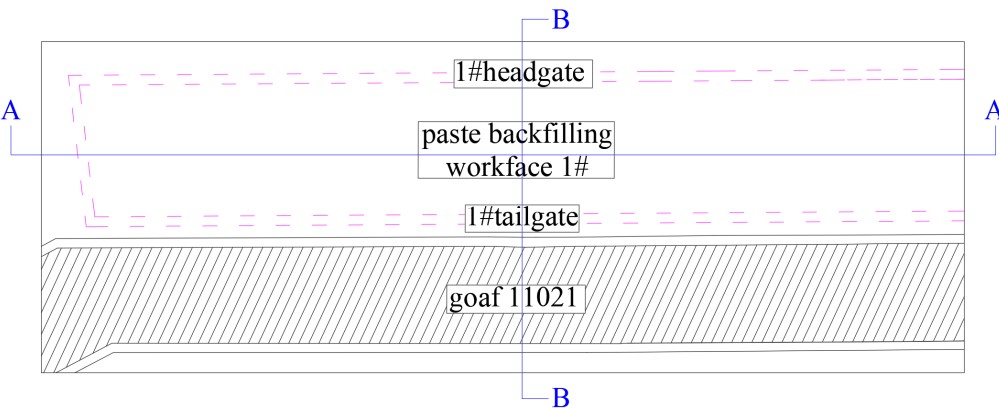

**Figure 5.** The panel layout of numerical simulation.

The model built was 250 m long, 220 m wide and 122 m high. The mining of panel 11021 has been finished for more than 20 years. Therefore, by weakening the coal and rock mass mechanical parameters of goaf 11021 in a certain proportion, the stress distribution of panel 1# under different backfilled ratios was simulated. The specific weakening parameters were as follows: shear modulus decreased by 85%, bulk modulus decreased by 85% and cohesion decreased by 90%. The average buried depth of the working face was 700 m and the overlying load was 17.5 MPa/m. The model was loaded according to the calculated weight of the rock stratum. After the software solution reached equilibrium, the initial equilibrium model of numerical simulation was obtained and is shown in Figure 6.

### 4.2. Analysis of Numerical Simulation Results

The vertical stress, vertical displacement and plastic zone distribution after 150 m of excavation of the panel were simulated and analyzed when the backfilled ratios were 0%, 80%, 90% and 96%, respectively (i.e., the equivalent mining height was 2.8 m, 0.56 m, 0.28 m and 0.11 m, respectively). Numerical simulation results and analysis are as follows.

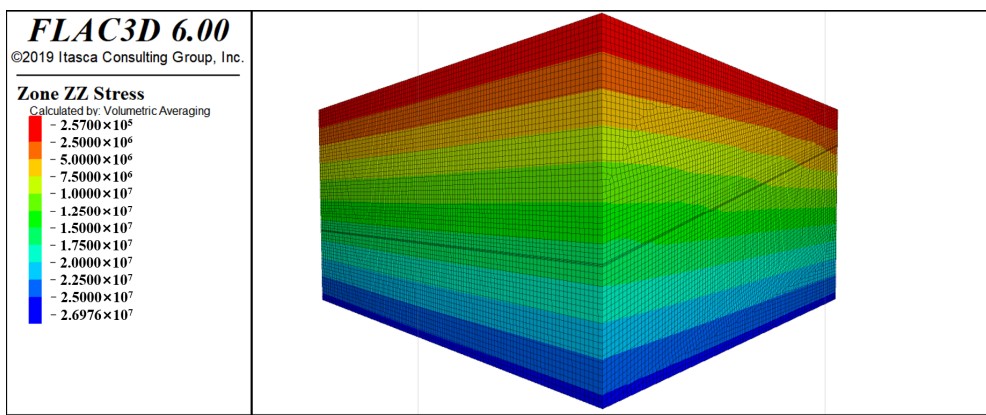

**Figure 6.** Numerical simulation initial equilibrium model of 1# working face.

### 4.2.1. Vertical Stress Analysis

Figure 7 shows the vertical stress distribution of the coal seam slice location and vertical slice location for different simulation schemes. Due to the low bearing capacity of the caving strata in the goaf 11021 and the yielding of the 5m wide barrier pillar, the vertical stress concentration degree and range in the headgate side under different backfilled ratios were significantly larger than those in the tailgate side. After the excavation of the panel, the backfilling body contacted with the sinking roof and supported it. As the roof subsidence in the middle of the goaf was the largest, the supporting effect of the backfilling body on the roof gradually weakened from the middle of the goaf to the surrounding area.

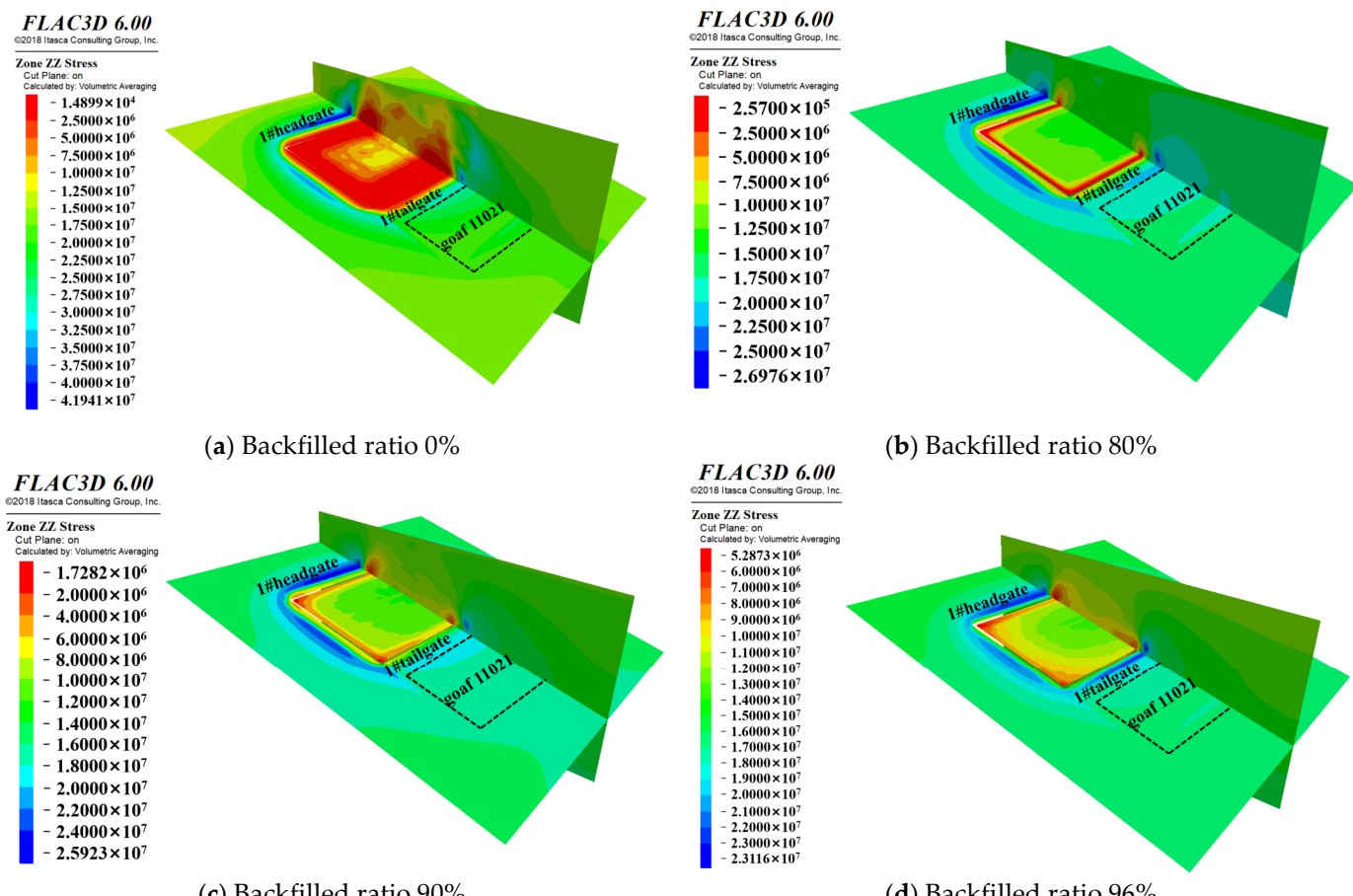

**Figure 7.** Vertical stress distribution of the panel 1#.

Figure 8a shows the vertical stress curve along the tendency of the panel. It can be seen from the figure that with the increase in the backfilled ratio of the panel, the peak stress of the coal beside the headgate decreased from 36.8 MPa to 21.1 MPa and the stress concentration coefficient decreased from 2.3 to 1.32, while that beside the tailgate decreased from 27.1 MPa to 19.5 MPa and from 1.69 to 1.22. The stress concentration coefficient of the coal beside the headgate and the tailgate reduced by 43% and 28%, respectively, which indicated that backfilling mining can significantly reduce the stress concentration degree. With the increase in backfilled ratio, the peak stress difference between the two roadways gradually decreased. When the backfilled ratio reached 96%, the stress concentration coefficient of the two gates was the same and the peak stress was basically equal. At this time, the goaf was supported by the backfilling body and the overall stress state was basically the same as that before mining; therefore, the stability of the roadway was greatly improved.

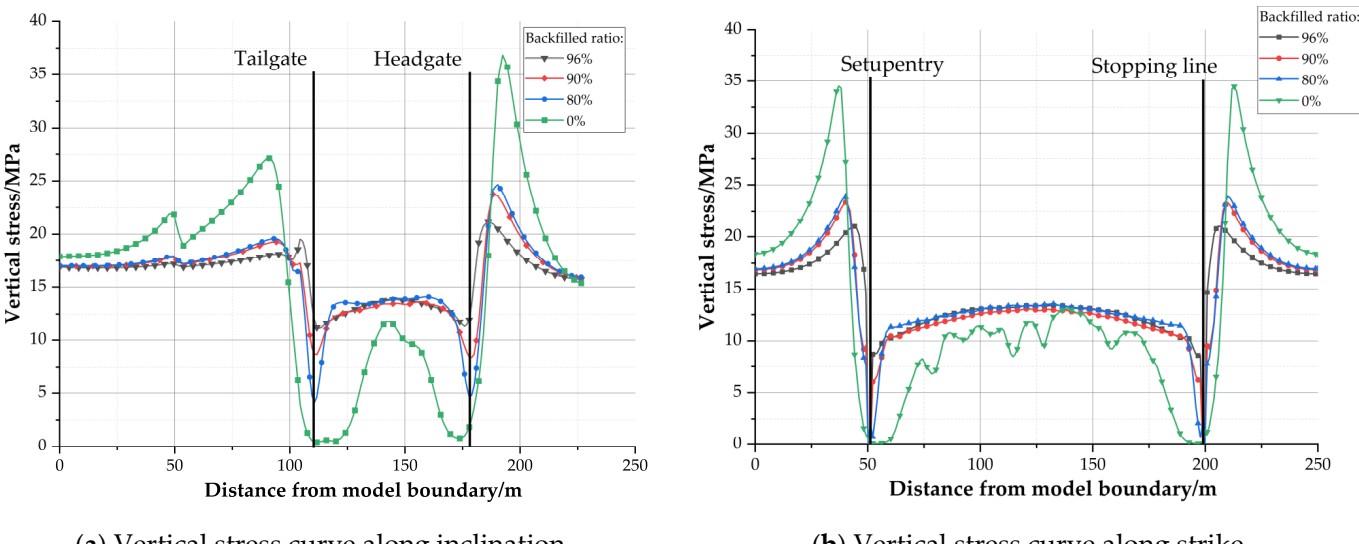

(**a**) Vertical stress curve along inclination  (**b**) Vertical stress curve along strike

**Figure 8.** Vertical stress curve at different filling rates.

Taking 1.1 times of gravity stress as the critical stress, when the stress in the coal in front of the working face is lower than the value, it is regarded as not affected by the front abutment pressure, so as to determine the influence range of the front abutment pressure. It can be seen from Figure 8b that with the increase in the backfilled ratio of the goaf, the supporting effect of the backfilling body on the overlying strata was continuously enhanced. The peak concentration coefficient and the influence range of the front abutment pressure of the working face was reduced from 2.16 to 1.31 and 60 m to 23 m, respectively.

According to the theory of dynamic and static combined load [24], the higher the static load on the coal body, the easier it is to destabilize the damage. Therefore, the peak stress in front of the working face under different backfilled rates in the simulation results is selected for curve fitting, and the polynomial fitting method with the smallest error is selected. The relationship between peak stress and filling rate is as follows:

$$\sigma_{max} = 2.33\eta^2 - 18.5\eta + 36.84 \tag{3}$$

where $\sigma_{max}$ is the peak stress and $\eta$ is the backfilled ratio.

The derivation of Equation (3) shows that as the backfilled ratio increases, the peak stress decreases, but the peak stress reduction rate also decreases, that is, the larger the backfilled ratio, the weaker the improvement effect of backfilling mining.

### 4.2.2. Vertical Displacement Analysis

Figure 9 shows the vertical displacement along the tendency of the goaf under different backfilled ratios. The simulation results showed that as the backfilled ratio of the goaf increased from 0% to 96%, the maximum vertical displacement of the overburden gradually decreased from 2.75 m to 0.225 m, and the range of overlying strata with displacements also decreased, which indicates that the backfilled ratio is the key to controlling the overlying rock movement, and the larger the backfilled ratio, the smaller the overlying rock movement.

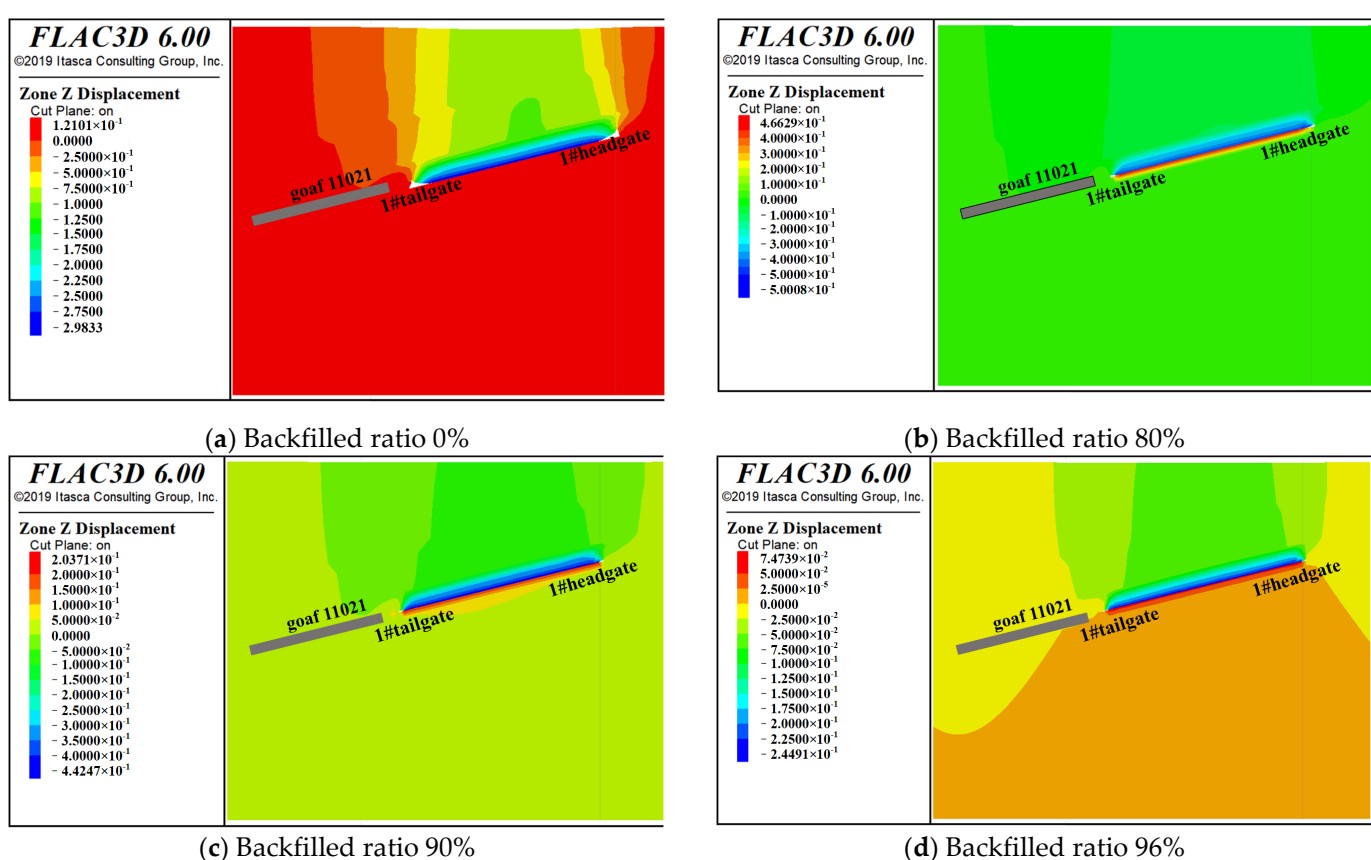

(**a**) Backfilled ratio 0%          (**b**) Backfilled ratio 80%

(**c**) Backfilled ratio 90%          (**d**) Backfilled ratio 96%

**Figure 9.** Vertical displacement along working face inclination at different backfilled ratios.

### 4.2.3. Plasticity Zone Analysis

Figure 10 shows the distribution of the plastic zone along the tendency of the goaf under different backfilled ratios. The simulation results showed that the development height of the roof plastic zone was 24 m when there was no backfilling, 5 m when the backfilled ratio was 80%, 4 m when the backfilled ratio was 90%, and 3 m when the backfilled ratio was 96%. With the increase in the backfilled ratio, backfilling mining can greatly reduce the development height of the roof plastic zone, and weaken the roof breaking degree and the dynamic ground pressure strength.

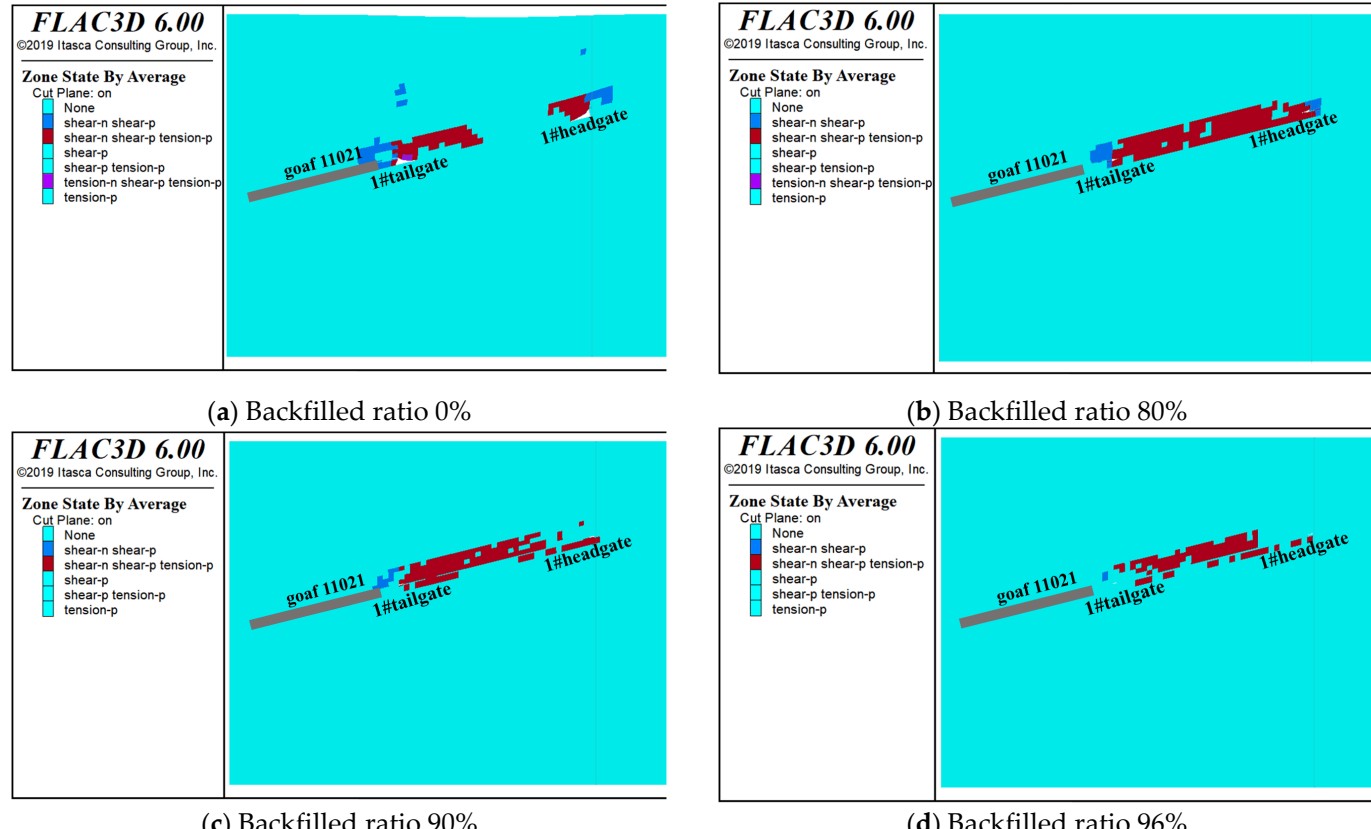

(**a**) Backfilled ratio 0%

(**b**) Backfilled ratio 80%

(**c**) Backfilled ratio 90%

(**d**) Backfilled ratio 96%

**Figure 10.** Distribution of goaf plastic zone along inclination under different backfilled ratios.

## 5. Analysis of Field Data and Load Reduction Effect of Paste Backfilling Workface 1#

*Analysis of Stress Monitoring Data of Two Roadways of Paste Backfilling Workface 1#*

During the mining of paste backfilling workface 1#, 23 groups of stress detectors were laid out in the headgate and tailgate to monitor the front abutment pressure of the coal seam. The stress detectors were arranged 60 m ahead of the working face in the two roadways with 30 m distance between the groups, and two stress detectors were set up in one group and arranged on the mining side coal rib; the buried depths were 8 m and 15 m, respectively. Among them, 11 groups of stress detectors were arranged in the headgate and 12 groups in the tailgate. The distribution of stress detectors in paste backfilling workface 1# is shown in Figure 11.

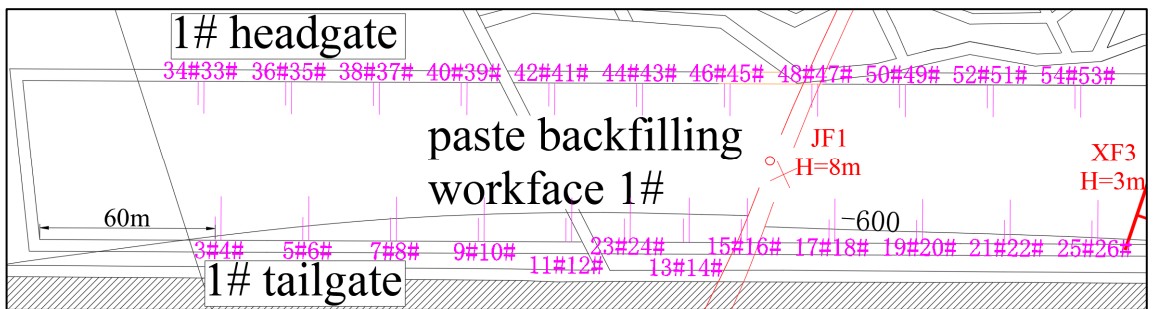

**Figure 11.** Layout of stress detectors in paste backfilling workface 1#.

Four consecutive groups of stress detector monitoring data were selected in each of the two roadways for analysis to obtain the distribution characteristics of front abutment pressure under backfilling mining conditions, as shown in Figures 12 and 13.

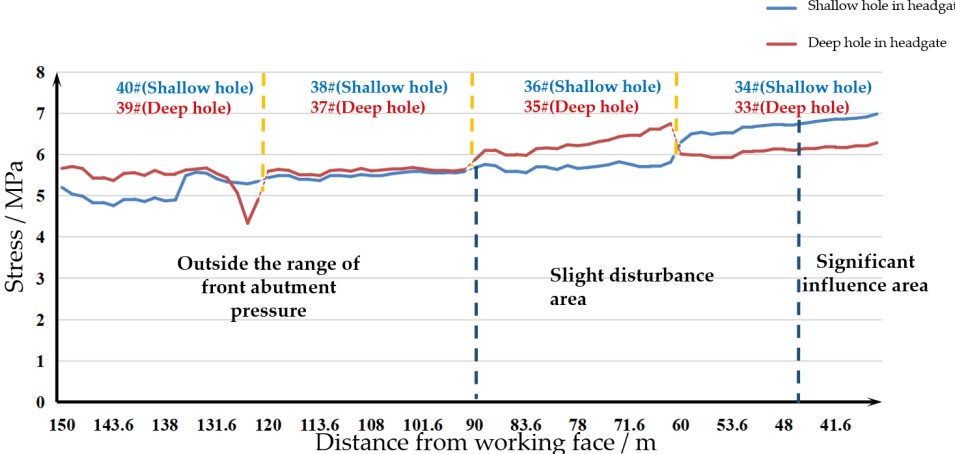

**Figure 12.** Front abutment pressure line chart of the headgate.

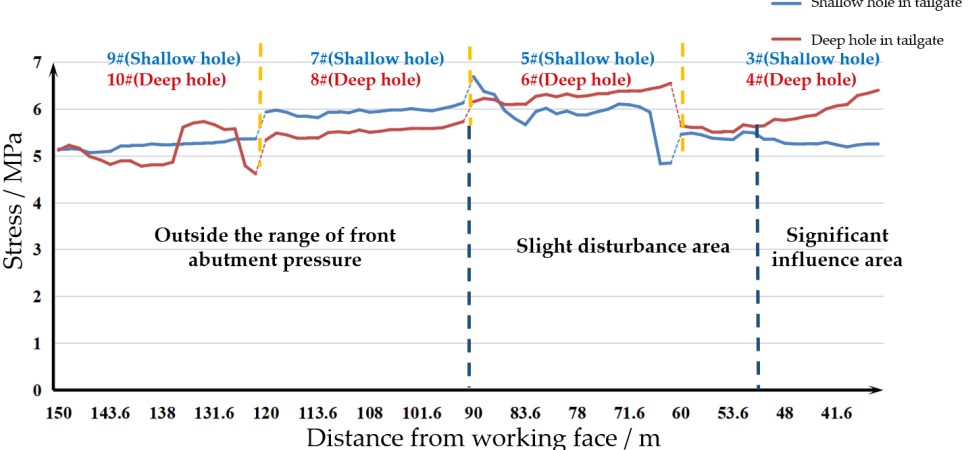

**Figure 13.** Front abutment pressure line chart of the tailgate.

The analysis results show that with the mining of the panel, the influence of the front abutment pressure on the two roadways can be divided into two stages: slight disturbance stage and significant influence stage. The slight disturbance stage lasted from about 90 m to 46 m from the working face, and the significant influence stage lasted from about 46 m to the working face. As the working face had several times of centralized pressure relief in the middle and tail fault areas of the two roadways in early September, the stress monitoring data in the middle of the two roadways fluctuated to some extent, but the overall front abutment pressure increased little under the support of the backfilling body. The front abutment pressure increment of the two roadways did not exceed 2 MPa during the mining process, which indicates that the support effect of backfilling body on the roof was very effective, and the front abutment pressure of headgate and tailgate was very small.

In summary, according to the stress monitoring results of the headgate and tailgate, the influence range of front abutment pressure of paste backfilling workface 1# was about 46 m, and the stress increment was not more than 2 Mpa.

In order to better illustrate the effect of backfilling mining in controlling the overburden fracture and reducing the dynamic ground pressure, a microseismic event comparison analysis was conducted between the paste backfilling workface 1# and the panel 1316 with similar geological conditions and mined using the caving method. The microseismic events distribution in paste backfilling workface 1# is shown in Figure 14.

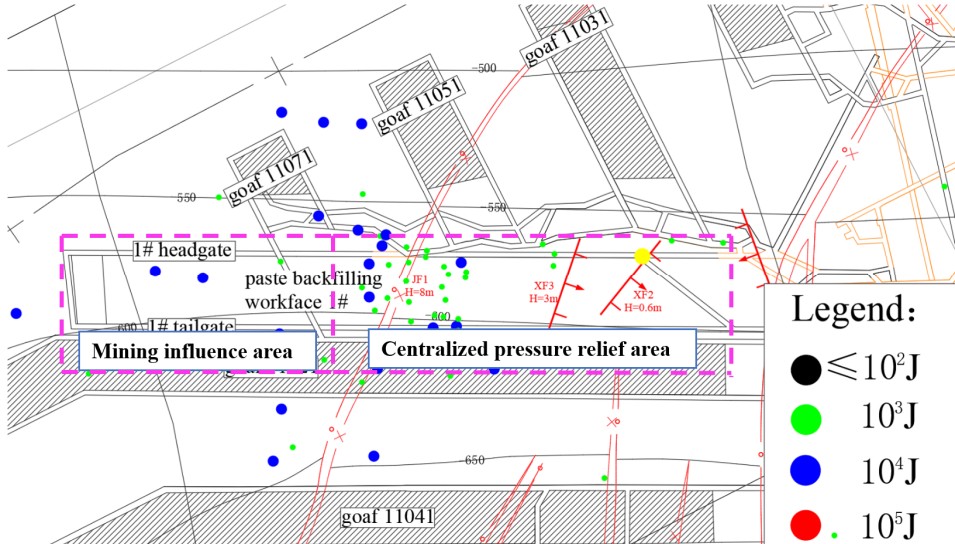

**Figure 14.** Microseismic events distribution in paste backfilling workface 1#.

The paste backfilling workface 1# advanced 31 m from 20 August to 16 October, and a total of 82 mine earthquake events were monitored. The distribution of microseismic events is shown in Figure 14. Among them, there were 21 events with a vibration energy at the level of 10 J, 34 events at the level of $10^2$ J, 26 events at the level of $10^3$ J and 1 event at the level of $10^4$ J. Among the above microseismic events, 58 were caused by the pressure relief construction, and the event at the level of $10^4$ J was caused by bottom coal blasting. Among the remaining 25 microseismic events, 15 were distributed in the surrounding goaf and coal pillar area. Therefore, there were only 10 microseismic events caused by the mining of paste backfilling workface 1#.

261 mine microseismic events were monitored when the panel 1316 mined using the caving method advanced 35 m, as shown in Figure 15. Among them, there was one event with a vibration energy at the level of 10 J, and there were 22 events at the level of $10^2$ J, 227 events at the level of $10^3$ J and 11 events at the level of $10^4$ J. The microseismic events distribution in panel 1316 is shown in Figure 15.

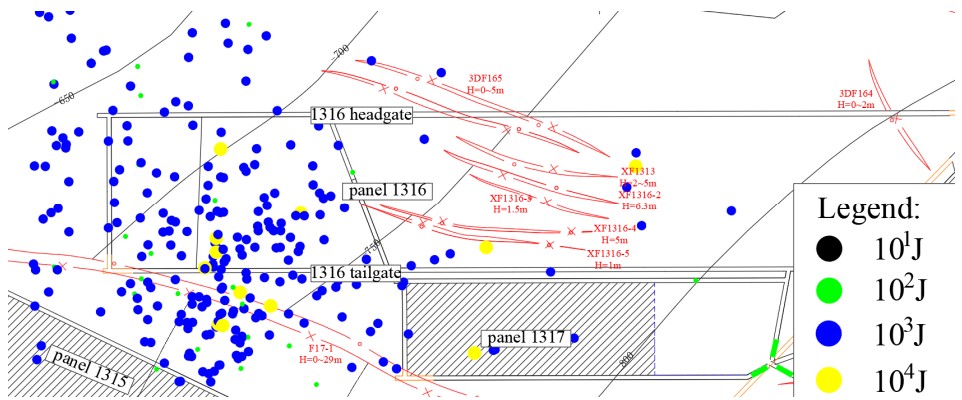

**Figure 15.** Microseismic events distribution in panel 1316.

It can be seen from Figures 14 and 15 that by the mining of paste backfilling workface 1#, there was a lower earthquake frequency and a lower energy level. Most of the mine earthquakes were caused by pressure relief construction in the fault area, and the roof activity caused by mining was very weak. On the other hand, the panel 1316 was mined using the caving method, and the microseismic events were mostly concentrated near the working face, with a high frequency and a relatively large energy level, indicating that the intensity of roof activity is significantly higher than that of the backfilling workface.

Figures 16 and 17 show the daily frequency and total daily energy variation in mine earthquakes monitored over 62 days in paste backfilling workface 1# and panel 1316. The results of the data analysis show that the maximum daily frequency of mine earthquakes was 12 and the average daily frequency of mine earthquakes was 1.34 during the 62 days of mining of paste backfilling workface 1#, while that of panel 1316 was 10 and 4.26, respectively. The maximum daily total energy of paste backfilling workface 1# was $4.78 \times 10^4$ J, and the average daily total energy was $1.80 \times 10^3$ J, while the maximum daily total energy of panel 1316 was $6.69 \times 10^4$ J and the average daily total energy was $1.76 \times 10^4$ J. Therefore, the average daily frequency and average daily total energy of mine earthquakes at paste backfilling workface 1# was only 31% of that at panel 1316.

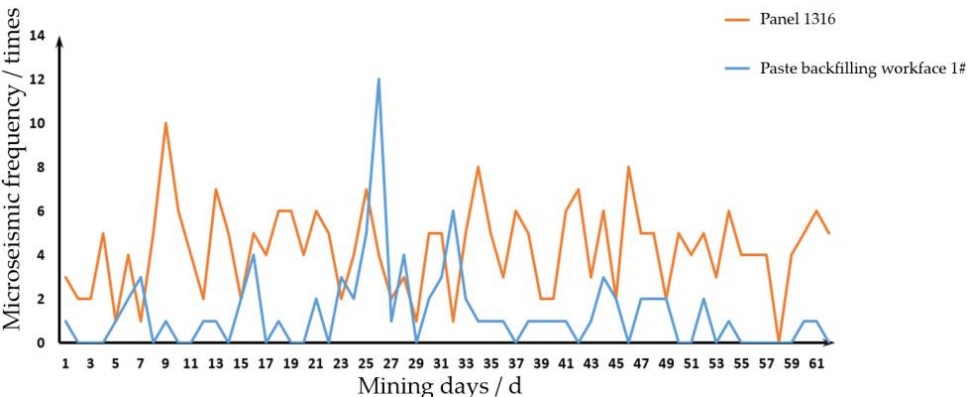

**Figure 16.** Comparative line graph of daily frequency of microseismic events.

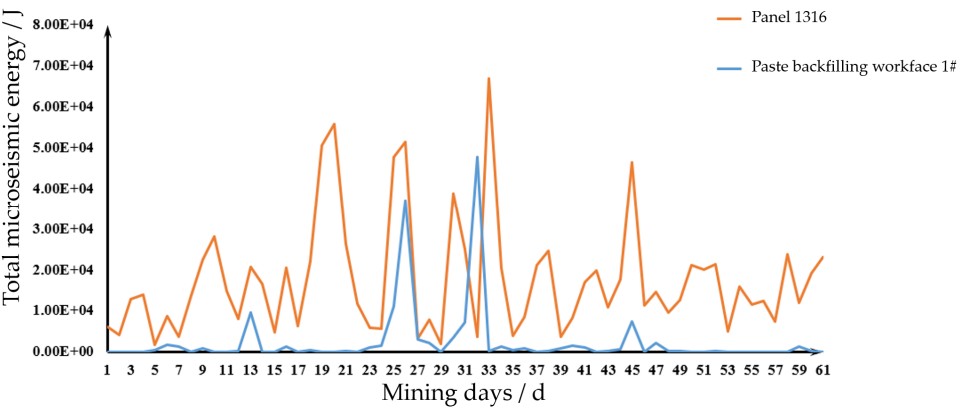

**Figure 17.** Comparative line graph of daily total energy of microseismic events.

In summary, whether it is the frequency or the energy of mine earthquakes, backfilling mining is far lower than caving mining. Backfilling mining greatly reduces the degree of overburden fracture and the potential danger of dynamic ground pressure.

## 6. Conclusions and Prospect

### 6.1. Conclusions

1.  Backfilling mining controls the movement and deformation of overburden by reducing the activity range of roof strata, and the backfilled ratio is the key to controlling the movement degree of overburden. The fracture development height of the overlying strata of the goaf decreases with the increase in the backfilled ratio. However, due to the roof deflection before backfilling and the defective distance of roof contact, the backfilled ratio of the panel cannot reach 100%; thus, there is a boundary effect.
2.  The numerical simulation results showed that with the increase in the backfilled ratio, the influence range of the front abutment pressure, the stress concentration degree

of the two roadways, the displacement of the overlying strata and the development height of the plastic zone were all reduced to varying degrees.

3. The analysis of stress monitoring data showed that the influence range of the front abutment pressure of the paste backfilling workface 1# was about 90 m and the significant influence range was about 46 m. The maximum stress of the two roadways did not exceed 7 MPa, while the influence range of the vertical stress and the front abutment pressure of the two roadways was small.

4. The analysis of microseismic events showed that the average daily frequency of microseism was 1.34, and the average daily total energy of microseism was $1.80 \times 10^3$ J, which decreased by 69% and 90%, respectively, compared with the caving method working face with similar geological conditions.

5. According to the result above, the field monitoring results are basically consistent with the theoretical analysis and numerical simulation results; thus, backfilling mining has a significant effect on controlling the movement of the roof strata and the ground pressure behavior.

### 6.2. Prospect

As resource development continues to move deeper into the earth, the risk of rock burst in coal mines around the world has gradually become prominent [23–25]. Since backfilling mining can effectively prevent rock burst and protect the surface environment, it will become one of the preferred means to solve the risk of rock burst in the future. In this paper, the law of overburden movement and mine pressure behavior under the condition of backfilling mining was studied, and the control effect of backfilling mining on rock burst was explored. However, the process and related technology of backfilling mining are not optimized. The main constraint to the widespread use of backfilling mining around the world is its high cost. Therefore, the direction of future research on filling mining should be to continuously optimize its related processes and technologies to ensure the filling effect while reducing production costs. If progress can be made in the above areas, filling mining will have good prospects for worldwide application.

**Author Contributions:** Conceptualization, J.L.; methodology, W.R.; validation, S.L. (Songyue Li), S.L. (Shun Liu) and W.R.; formal analysis, K.Y.; investigation, S.L. (Songyue Li); resources, H.L.; writing—original draft preparation, S.L. (Songyue Li); writing—review and editing, J.L.; supervision, J.L.; project administration, J.L.; funding acquisition, J.L. All authors have read and agreed to the published version of the manuscript.

**Funding:** This research was funded by the National Natural Science Foundation of China (U21A20110), the Research Fund of the State Key Laboratory of Coal Resources and Safe Mining, CUMT (SKL-CRSM22KF008), the Major Project of Natural Science Research in Colleges and Universities in Anhui Province (KJ2021ZD0051) and the National Natural Science Foundation of China (52004004).

**Institutional Review Board Statement:** Not applicable.

**Informed Consent Statement:** Not applicable.

**Data Availability Statement:** The data that support the findings of this study are available from the corresponding author upon reasonable request.

**Acknowledgments:** J.L. was supported by the National Natural Science Foundation of China (U21A20110), the Research Fund of the State Key Laboratory of Coal Resources and Safe Mining, CUMT (SKLCRSM22KF008), the Major Project of Natural Science Research in Colleges and Universities in Anhui Province (KJ2021ZD0051) and the National Natural Science Foundation of China (52004004).

**Conflicts of Interest:** The authors declare no conflict of interest.

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
