# Peer review of "Seismic Reduction Mechanism and Engineering Application of Paste Backfilling Mining in Deep Rock Burst Mines"

_sustainability, doi:10.3390/su15054336_

Round 1

Reviewer 1 Report

This scientific publication does not require changes in terms of methodology and calculation results

Author Response

Response to Reviewer 1 Comments

Point 1: This scientific publication does not require changes in terms of methodology and calculation results.

Response 1: Thank you for your valuable suggestions and affirmation for our study. I have made some changes to the grammar, style and words of the article, and made targeted changes to the article according to the opinions of other reviewers. We hope to receive your further comments and opinions.

Reviewer 2 Report

A very interesting article, but before publication I would recommend correcting minor comments:

- in Figure 1, the text is not clearly visible, it can be separated from each other by color;

- also in Figure 10-11 the text is not clear, it should be enlarged for clarity;

- on line 89 it says "Zhang et al. 2008", you can indicate in numbers and add below where the list of references is.

Author Response

Response to Reviewer 2 Comments

Point 1: In Figure 1, the text is not clearly visible, it can be separated from each other by color.

Response 1: Thank you for your comments. We have modified Figure 1 based on your comments to make the text in the figure clearer. Kindly refer to Figure 1 in the revised manuscript.

Point 2: Also in Figure 10-11 the text is not clear, it should be enlarged for clarity

Response 2: Thank you for your comments. We apologize for the negligence of our work. We also modified these figures to make the text clearer.

Point 3: On line 89 it says "Zhang et al. 2008", you can indicate in numbers and add below where the list of references is.

Response 3: Thank you very much. We have modified it and checked the full text to ensure that there are no other similar problems.

Reviewer 3 Report

The paper presented a work on the behavior of overburden movement and mine pressure of the paste backfilling workface. This topic can be interesting to readers in this journal. Some meaningful results were obtained, and the research can provide technical guidance and theoretical support for improving the safety of deep mining and the recovery rate of coal resources. I think the authors have done a good job and this paper can be published after revision. Here are some specific suggestions.

1. I strongly agree with the author 's point of view that the backfilling ratio is the key to control the ground pressure behavior of backfilling mining, but the author did not elaborate on the filling process of the backfilling workface and the detailed parameters of the backfill material in the article. It is suggested that the author supplement the above content.

2. The scale of the strip coal pillar is one of the key factors affecting its law of pressure behavior, but it is not explained in detail in the article. It is suggested that the author supplement the detailed parameters of the panel 1# and its surrounding working face.

3. The detailed stress detectors distribution position allows readers to have a clear and intuitive understanding of the 5.1 section. It is suggested to add the layout of stress detectors in working face in section 5.1.

4. Figure 8 and Figure 9 cannot intuitively reflect the change of stress distribution in working face under different backfilling ratio. It is suggested to merge the four subgraphs in Figure 8 and Figure 9 respectively.

5. The structure of the Section 4.2 “Analysis of numerical simulation results” is not clear enough. It is suggested to add a three-level title to each part of the analysis.

6. The author needs to explain in detail in the "Introduction" what is the current situation of coal gangue classification. Moreover, the significance and progress of your research needs to be explained in "introduction" section. Some references should be cited as follows: (a) Influence of backfilling rate on the stability of the “backfilling body-immediate roof” cooperative bearing structure. Int J Min Sci Technol 2022. (b) Investigation of collector mixtures on the flotation dynamics of low-rank coal. Fuel 2022; 327:125171.

7. The “Conclusions” section should be refined and shorten.

8. The quality of the figures should be improved.

9. Authors should carefully check the format of references and citations.

Author Response

Response to Reviewer 3 Comments

Point 1: I strongly agree with the author 's point of view that the backfilling ratio is the key to control the ground pressure behavior of backfilling mining, but the author did not elaborate on the filling process of the backfilling workface and the detailed parameters of the backfill material in the article. It is suggested that the author supplement the above content.

Response 1: Thank you for your valuable suggestions and affirmation for our study. We strongly agreed upon your suggestions. Therefore, we have added detailed description of backfilling process and materials in Sections 3.1 and 3.2 respectively.

Point 2: The scale of the strip coal pillar is one of the key factors affecting its law of pressure behavior, but it is not explained in detail in the article. It is suggested that the author supplement the detailed parameters of the panel 1# and its surrounding working face.

Response 2: Your advice is very pertinent, and we apologize that we did not take this into account before. We have supplemented the detailed parameters of the adjacent goaf and the ratio of mined to unmined width in Section 3.1.

Point 3: The detailed stress detectors distribution position allows readers to have a clear and intuitive understanding of the 5.1 section. It is suggested to add the layout of stress detectors in working face in section 5.1.

Response 3: Thank you for your advise. We have supplemented a layout figure and detaied description of stress detectors at the beginning of Section 5.1.

Point 4: Figure 8 and Figure 9 cannot intuitively reflect the change of stress distribution in working face under different backfilling ratio. It is suggested to merge the four subgraphs in Figure 8 and Figure 9 respectively.

Response 4: Thank you for your advise. We have rearranged the results of numerical simulation and merged the Figure 8 and Figure 9 respectively. Kindly refer to the Figure 8 in the revised manuscript.

Point 5: The structure of the Section 4.2 “Analysis of numerical simulation results” is not clear enough. It is suggested to add a three-level title to each part of the analysis.

Response 5: We agreed upon your comments. Therefore we categorized the content in Section 4.2 and add a third-level title to each part of the analysis to make its structure clearer.

Point 6: The author needs to explain in detail in the "Introduction" what is the current situation of coal gangue classification. Moreover, the significance and progress of your research needs to be explained in "introduction" section. Some references should be cited as follows: (a) Influence of backfilling rate on the stability of the “backfilling body-immediate roof” cooperative bearing structure. Int J Min Sci Technol 2022. (b) Investigation of collector mixtures on the flotation dynamics of low-rank coal. Fuel 2022; 327:125171.

Response 6: Thank you for your valuable suggestions. We have carefully reviewed the research in the field of coal gangue classification and studied the references you recommend. The current situation and some research results of coal gangue classification have been summarized in the introduction, and the references you recommend has been cited. Besides, the significance of our research has been supplemented in the last paragraph of the Introduction. Kindly refer to the Introduction in the revised manuscript.

Point 7: The “Conclusions” section should be refined and shorten.

Response 7: Thank you very much. We strongly agreed with the suggested revisions. After discussion, we delete the unnecessary part of the conclusion and refine it into five parts.

Point 8: The quality of the figures should be improved.

Response 8: Thank you for your advise. Several other reviewers also mentioned this problem. We attach great importance to this and have modified or redrawn some of the figures. The figures involved in the modification are Figure 1, Figure 2, Figure 8 to 10, Figure 12 to 13 and Figure 16 to 17.

Point 9: Authors should carefully check the format of references and citations.

Response 9: Thanks for your reminder, we carefully checked the citations and references, modified some format errors, and added DOI to the references.

Reviewer 4 Report

The article of the authors is devoted to an important and topical issue related to the issues of reducing seismic activity and the use of backfilling of mining operations in mines. The importance of the mining industry as a huge branch of the national economy associated with the extraction and processing of mineral raw materials will continue for the coming decades. Increasing demand for various types of mineral raw materials leads to the deepening of mining operations at exploited deposits and the involvement in the development of deposits that are unfavorable in terms of climatic or mining and geological conditions, and a decrease in the cut-off content of useful components causes an increase in the volume of extracted and transported rock mass. All this leads to a violation of the geodynamic regime of the geophysical environment, which entails a decrease in the technical and economic efficiency and safety of mining operations. The studies presented in the paper are undoubtedly of interest to readers in the field under consideration.

 However, it would be necessary to clarify a number of comments that are available to the article:

1. The data shown in Figure 2 for the Gucheng coal mine is interesting. Has the presented face paste scheme been compared with similar coal mines in other regions of the world?

2. It would be necessary to present the obtained regression models of the dependence of the overburden opening height on the backfill coefficients and the corresponding coefficients of determination.

3. The article should have disclosed in more detail how the numerical simulation of the initial equilibrium face model was carried out, the results of which are presented in Figure 5.

4. The results presented in Figures 8, 9 should have been supplemented with mathematical models that could be used to determine the optimal values of the calculated parameters under various conditions.

5. It is not entirely clear from the article whether it is planned to issue a patent for the analysis technique used and to what extent it can be tested in similar conditions?

6. The research methods used by the authors could later be applied to similar coal mines, in particular, to European conditions. In particular, one could consider the article: Bosikov I. I., Klyuev R. V., Khetagurov V. N. Analysis and comprehensive evaluation of gas-dynamic processes in coal mines using the methods of the theory of probability and math statistics analysis. Sustainable Development of Mountain Territories. 2022;14(3):461-467. DOI: 10.21177/1998-4502-2022-14-3-461-467.

7. We should dwell on further prospects for conducting these interesting studies, including taking into account the new realities in the world.

Author Response

Response to Reviewer 4 Comments

Point 1: The data shown in Figure 2 for the Gucheng coal mine is interesting. Has the presented face paste scheme been compared with similar coal mines in other regions of the world.

Response 1: In the early stage of the working face design, we compared the paste backfilling schemes with other similar coal mines in the world. However, limited by the space of the article, we did not show this part of the work in the first draft. In this manuscript, we added a brief description of the selection of filling materials.

Point 2: It would be necessary to present the obtained regression models of the dependence of the overburden opening height on the backfill coefficients and the corresponding coefficients of determination.

Response 2: Thank you for your advise. The mathematical model of overburden fracture development height was shown as Equation 2. The calculation of overburden fracture development height in Section3.3 was a theoretical prediction based on the Equation 2.

 Point 3: The article should have disclosed in more detail how the numerical simulation of the initial equilibrium face model was carried out, the results of which are presented in Figure 5.

Response 3: After discussions, we strongly agreed upon your suggestions. We have rearranged the content in Section 4.1 and illustrated how the initial equilibrium face model was carried out in detail. Kindly refer to the Section 4.1 in the revised manuscript.

Point 4: The results presented in Figures 8, 9 should have been supplemented with mathematical models that could be used to determine the optimal values of the calculated parameters under various conditions.

Response 4: Thank you for your advise. We selected the peak stress in front of the working face in the numerical simulation results for polynomial fitting, and obtained the mathematical model of peak stress and filling rate, and made a simple analysis. Kindly refer to the Section 4.2.1 in the revised manuscript.

Point 5: It is not entirely clear from the article whether it is planned to issue a patent for the analysis technique used and to what extent it can be tested in similar conditions.

Response 5: We apologize that we didn 't mention these issues in the article, since we are not intend to apply for a patent for the analysis method or technology involved in the article. These analysis methods are relatively simple and universal, and can theoretically be applied to most mines equipped with microseismic monitoring systems or stress monitoring systems.

Point 6: The research methods used by the authors could later be applied to similar coal mines, in particular, to European conditions.In particular, one could consider the article: Bosikov I. I., Klyuev R. V., Khetagurov V. N. Analysis and comprehensive evaluation of gas-dynamic processes in coal mines using the methods of the theory of probability and math statistics analysis. Sustainable Development of Mountain Territories. 2022;14(3):461-467. DOI: 10.21177/1998-4502-2022-14-3-461-467.

Response 6: Thank you for your approbation of our work. We have carefully studied the artical you recommend and have been inspired greatly. We have cited it as one of the references.However,we are sorry that it is difficult for us to make further analysis of the similar coal mines in Europe due to the limitation of space and structure of our article. But we have dwelled on prospects of backfilling mining and took into account the new realities in the world. Kindly refer to the Section 6.2 in the revised manuscript.

Point 7: We should dwell on further prospects for conducting these interesting studies, including taking into account the new realities in the world.

Response 7: We strongly agreed upon your suggestions. Therefore, we have discessed the main constraint to the widespread use of backfilling mining and its research directions in the future in Section 6.1.

Round 2

Reviewer 3 Report

The paper has been well modified and can be accepted.